# Extraction, Purification, Characterization and Application in Livestock Wastewater of S Sulfur Convertase

**DOI:** 10.3390/ijerph192316368

**Published:** 2022-12-06

**Authors:** Xintian Li, Wei Zhai, Xinran Duan, Changlong Gou, Min Li, Lixia Wang, Wangdui Basang, Yanbin Zhu, Yunhang Gao

**Affiliations:** 1College of Animal Science and Technology, Jilin Agricultural University, Changchun 130000, China; 2College of Animal Science and Technology, Inner Mongolia University for Nationalities, Tongliao 028000, China; 3Northeast Institute of Geography and Agroecology, Chinese Academy of Sciences, Changchun 130102, China; 4Institute of Animal Husbandry and Veterinary Medicine, Tibet Academy of Agricultural and Animal Husbandry Science, Lhasa 850009, China

**Keywords:** sulfide, extracellular enzyme, purification, enzymatic property, livestock wastewater

## Abstract

Sulfide is a toxic pollutant in the farming environment. Microbial removal of sulfide always faces various biochemical challenges, and the application of enzymes for agricultural environmental remediation has promising prospects. In this study, a strain of Cellulosimicrobium sp. was isolated: numbered strain L1. Strain L1 can transform S^2−^, extracellular enzymes play a major role in this process. Next, the extracellular enzyme was purified, and the molecular weight of the purified sulfur convertase was about 70 kDa. The sulfur convertase is an oxidase with thermal and storage stability, and the inhibitor and organic solvent have little effect on its activity. In livestock wastewater, the sulfur convertase can completely remove S^2−^. In summary, this study developed a sulfur convertase and provides a basis for the application in environmental remediation.

## 1. Introduction

Sulfide is one of the main environmental pollutants in wastewater from livestock and aquaculture. The main component of the unpleasant odor produced in poultry farms is H_2_S [1], which is produced by the decomposition sulfide of aerobic/anaerobic bacteria in livestock wastewater [2,3]. H_2_S can adversely affect organisms, such as poultry chronically exposed to >10 ppm H_2_S, and it can cause intestinal inflammation and reduce egg production [4,5]. Therefore, it is necessary to treat sulfide pollution in livestock wastewater.

Microbial removal has received extensive attention as an efficient and economical method. Microorganisms can convert harmful sulfur compounds into harmless products. The previous study has examined the potential for biofiltering with aerobic, chemotrophic [6], and phototrophic microbial consortia for the removal of sulfide from biogas [7]. However, 30–250 mg/L sulfide can inhibit the growth of microorganisms [8,9], and the sulfide transformation process by microorganisms is very time consuming in sewage [10,11]. Biological processes (such as activated sludge and trickling filters) are considered more beneficial than microbial processes because of their effectiveness; however, most of the emerging contaminants remain soluble in wastewater and cannot be eliminated [12]. The main advantage of the above methods is the production of various intracellular and extracellular enzymes that can degrade contaminants [13]. Similarly, the removal of sulfide by microorganisms relies heavily on the combined action of enzymes in the organism. Therefore, an exploration of the application of enzymes to treat pollutants in wastewater was conducted [14].

For example, lysozyme treatment of microorganisms causes the bacteria to release intracellular substances, thus improving the sewage treatment [15]. Garbage enzyme (GE), an organic solution rich in enzymes, has been applied in wastewater treatment [16]. Quinone oxidoreductase carbon matrix treats sulfide-rich tanning wastewater, and the removal rate of sulfide can reach 99% [17]. However, the application of enzymes to sulfide treatment in livestock wastewater is less studied. Based on this, the study was launched. In this study, the sulfur convertase was obtained and showed positive effects in livestock wastewater applications.

## 2. Materials and Methods

### 2.1. Enrichment and Isolation of Sulfur-Transforming Bacteria

To obtain sulfur conversion bacteria, wastewater (solid–liquid, separated and unfermented fresh wastewater, was collected from Guangze ecological pasture in Changchun, China) as a separation source. Next, 1 mL wastewater was added to 100 mL inorganic salt medium (NH_4_Cl 1 g/L, KH_2_PO_4_ 0.5 g/L, K_2_HPO_4_ 1.5 g/L, Na_2_HCO_3_ 0.1 g/L, NaCl 1 g/L, MgCl_2_ 0.2 g/L, sucrose 0.5 g/L), Na_2_S (purity: ≥98%; 500 g; CAS: 1313-84-4; Macklin; Shanghai, China) was used as a sulfur source. It was cultivated 5 times, and S^2−^ concentration was set to 100, 300, 500, 700, and 1000 mg/L; each time was set to 7 days. The microbial suspension was cultured on the LB (10.0 g/L tryptone, 10.0 g/L NaCl, and 5.0 g/L yeast extract) plate. Colonies were picked and streaked on fresh LB plates 3 times, respectively, transferring the colonies to 5 mL LB, cultured 24 h. Next, the bacteria were transferred to an inorganic salt medium to assess the S^2^ conversion capacity. Based on conversion results, strains were selected for subsequent experiments.

### 2.2. Morphology and Identification of the Strain L1

To observe the colony morphology of bacteria, the cultured bacteria were streaked on the LB plate until single colonies appeared. The colony was picked for gram staining and observed in an optical microscope (PH100-3B41L-IPL, Phenix, Jiangxi, China). Next, selected colonies were added to LB culture for 24 h, and DNA was extracted from the cultured strain. The extracted DNA was used as a template for 16S rRNA amplification and submitted to Sangon Biotech (Shanghai, China) for sequencing. The obtained sequence was aligned with existing sequences in NCBI. After the alignment, the sequences with high scores were selected to construct the phylogenetic tree (MEGA X was used).

### 2.3. Location and Validation of S^2−^ Transformation Active Components in the Strain L1

#### 2.3.1. Location of S^2−^ Transformation Active Components in Strain L1

To locate the main active components of the strain L1, sulfur conversion experiments were performed in the following groups: bacterial cultures (B cultures), bacterial cells (B cells), culture supernatant (CS, extracellular enzyme), and cell lysate (intracellular enzyme) of the strain L1 [18]. In brief, the strain L1 was incubated in the inorganic salt medium at 37 °C for 24 h; the obtained cultures were divided into 3 portions, with the first portion as B cultures. The second portion was centrifuged; the pellets were collected and washed 3 times with phosphate-buffered saline (PBS) solution as B cells. The third portion was centrifuged, and the supernatant was filtered through a sterile 0.22 um filter (JINTENG, Tianjin, China). The pellets were crushed on ice with an ultrasound cell crusher (VCX130PB, Sonics, Shanghai, China). After crushing, the supernatant was centrifuged and filtered to obtain cell lysate. S^2−^ solution was added to 4 groups, respectively, (S^2−^ final concentration set to 500 mg/L) for sulfur conversion experiments, with inorganic salt medium and PBS acting as controls [19].

#### 2.3.2. Effects of Heat, SDS, and Proteinase K Treatments on S^2−^ Conversion Activity by the Culture Supernatant of Strain L1

To verify the role of the L1 active site, the effect of proteinase K (PK), SDS, PK +SDS, and heat treatment on the S^2−^ conversion of CS was investigated [18]. CS was exposed to 2 mg/mL PK, 50 mg/mL SDS, and both simultaneous treatments and incubated at 4 °C for 6 h each to investigate the effect on S^2−^ conversion. The CS was boiled at 100 °C for 10 min and 1 h to explore the effect of heat treatment on S^2−^ conversion. The PBS and untreated culture supernatants were used as controls, respectively.

### 2.4. Optimization of Enzyme Conversion Ability

To improve the conversion ability of the extracellular enzyme, the medium composition was optimized by single-factor experiments. The culture components were optimized using various carbon sources (Sucrose, glucose, CH_3_COONa, sodium citrate, mannitol) and nitrogen sources (NH_4_Cl, NH_4_H_2_PO_4_, ammonium tartrate, NaNO_3_, Urea, yeast). After the carbon and nitrogen source selection was completed, the appropriate concentration was explored. CS was prepared by medium with changed conditions, and after completion of the preparation, sulfur conversion experiments were performed, and the appropriate component was chosen based on conversion results.

### 2.5. Extraction and Purification of the Sulfur Convertase

#### 2.5.1. Extraction and Purification of Sulfur Convertase

First, we prepared CS (crude enzyme), which was passed through a sterile 0.22 um filter (to completely remove bacterial interference). The beaker containing the crude enzyme solution was heated to 4 °C, and pre-ground (NH_4_)_2_SO_4_ powder was slowly added to make the solution saturation reach 20%, 40%, 60%, and 80% in turn. The protein content and enzyme activity of the precipitates were determined. The optimum precipitation interval S1 to S2 was calculated based on the enzymatic conversion activity. The enzyme solution obtained in the optimal precipitation interval was added to the activated 14 KD MW membrane and stood at 4 °C in PBS for 12 h, changing the PBS every 4 h to completely remove (NH_4_)_2_SO_4_ interference. The enzyme solution was concentrated by PEG20000, and the enzyme solution was further purified via SephadexG-75 gel filtration chromatography. The column was pre-equilibrated with PBS-NaN_3_ buffer, and the eluate was adjusted to a flow rate of 0.5 mL/min. After elution, the collection volume reached 5 mL, and the protein concentration and enzyme conversion activity were determined.

#### 2.5.2. Determination of Protein Concentration and Molecular Weight

Protein concentration was determined by the BCA method. A Pierce BCA protein assay kit was purchased from Thermo Fisher Scientific (Rockford, IL, USA). The molecular weights at different stages of purification were estimated using SDS-PAGE [20]. Proteins were separated with 12% (*w*/*v*) acrylamide, and a protein marker mixture (Beyotime, Shanghai, China) was applied to calibrate the molecular weight. We performed staining with 0.25% Coomassie blue (R-250) and de-staining with 1% acetic acid, observing results with a gel documentation system (GenoSens 1880, Clinx Science Instruments Co., Ltd., Shanghai, China).

### 2.6. Effect of Physiochemical Factors on Sulfur Convertase Activity

#### 2.6.1. Optimum Reaction Temperature and Thermal Stability of Enzymes

To explore the optimum reaction temperature for sulfur convertase, sulfur conversion experiments were conducted at 10–80 °C; the reaction time was 6 h.

To investigate the thermal stability of the enzyme, the enzymes were treated at different temperatures (10 °C, 20 °C, 30 °C, 40 °C, 50 °C, 60 °C, 70 °C, and 80 °C) for 2 h before the reaction, and then the sulfur conversion experiment was performed. The reaction was performed at 37 °C for 24 h.

#### 2.6.2. Stability of Sulfur Convertase over Storage Time

The purified sulfur convertase was stored at 4 °C for 30 days, and the enzyme was taken out every 3 days for sulfur conversion experiments to detect the change in the activity [21].

#### 2.6.3. Effect of Inhibitors and Organic Solvents on Enzyme Activity

To investigate the effect of inhibitors and organic solvents on enzyme activity, inhibitors including 0.1 and 1 mM PMSF, NaN_3_, 10, 50, and 100 mM EDTA; organic solvents including 10%, 20%, and 30% (*v*/*v*) methanol, ethanol, isopropanol, and DMSO; and the enzymes were treated with the above reagents separately for 2 h at 4 °C. After 2 h, S^2−^ was added for conversion experiments.

#### 2.6.4. Effect of Redox on Enzyme Activity

S^2−^ conversion is an oxidative reaction, so the sulfur convertase may become an oxidative enzyme, followed by addition of reductant to verify this conjecture. The enzymes were treated with 1 mmol/L dithiothreitol (DTT), 1 mmol/L ascorbic acid, and 10 mmol/L *β*-mercaptoethanol (sulfhydryl protectors) for 2 h at 4 °C. Afterwards, S^2−^ was added and the enzyme activity was measured after 24 h.

#### 2.6.5. Zymolyte Competition Experiments

To verify the effect of different substrates on the S^2−^ conversion, different concentrations of S_2_O_3_^2−^ (100, 500 mg/L) and SO_4_^2−^ (100, 500 mg/L) were added, and the enzyme activity was measured after 24 h. S_2_O_3_^2−^ concentration was determined by an improved iodometric method [22], and SO_4_^2−^ was determined by barium chromate spectrophotometry [23].

### 2.7. Sulfur Conversion Capacity of Enzymes under Livestock Wastewater

Sulfur conversion experiments were conducted in livestock wastewater (solid–liquid separated and unfermented fresh wastewater, collected from Guangze ecological pasture Changchun, China) to assess the availability of the sulfur convertase application. The experimental groups were as follows: group 1–strain L1, group 2¬¬¬–sulfur convertase. Each group added 1 mL to 100 mL livestock wastewater. The solution without addition of bacteria or enzymes was taken as the control. S^2−^ was not added in the experimental group due to the S^2−^ being present in the wastewater (75.7 ± 3.1 mg/L). The application ability of the sulfur convertase was evaluated by the changes in S^2−^ concentration, ammonia nitrogen, total phosphorus, and COD. The water-quality indicators were measured by a multiparameter water-quality meter (TR6900, Shenzhen Tongao Co., Ltd., Shenzhen, China). COD was determined by the dichromate titration [24]; the ammonia nitrogen was determined by Nessler’s reagent spectrophotometry [25]; and the total phosphorus was determined by the spectrophotometric method of ammonium molybdate [26].

### 2.8. Detection and Analysis Methods

Unless otherwise specified, the above sulfur conversion was conducted in a 2 mL EP tube, and S^2−^ concentration was set to 500 mg/L, in triplicate. After the reaction, the enzyme activity was calculated based on the S^2−^ concentration. S^2−^ was determined by p-aminodimethylaniline spectrophotometry [27]. The graphs were prepared using GraphPad Prism (version 8.0, GraphPad: San Diego, CA, USA), data statistical analysis with SPSS software (applying one-way ANOVA, F-test, and Least Significant Difference test; version 25.0, SPSS: Armonk, NY, USA).

## 3. Results and Discussion

### 3.1. Isolation and Identification of Sulfur-Transforming Strain L1

Among the 12 bacterial strains isolated from wastewater, strain L1 showed the highest sulfide removal efficiency (Table 1). The control group also showed a 14.5% conversion rate, which may be due to the reaction of S^2−^ with oxygen in the air. Therefore, strain L1 was chosen for further characterization. The colonies of strain L1 were yellow, round, and raised, with neat edges (Figure 1a). The strain was a short rod-shaped Gram-positive bacterium (Figure 1b).

Based on the gene sequence comparison, strain L1 showed the highest similarity to *Cellulosimicrobium* sp. (Figure 2), and the accession number (MZ687074) was submitted to GenBank. Previous studies showed that the *Cellulosimicrobium* sp. isolated from soil, marine sediment, and sewage can degrade 2,4,5-trichlorophenoxyacetic acid and biodiesel-oil [28,29,30]; however, studies on inorganic sulfur conversion are scarce. Therefore, further exploration was conducted.

### 3.2. Location of Active Component of S^2−^ Conversion by Strain Cellulosimicrobium sp. L1

The results of transforming sulfur with different components of strain L1 are shown in Figure 3a. The S^2−^ conversion rate of CS (60.37%) was higher than that of the other components after 24 h reaction. This shows that some extracellular substances of strain L1 play a significant role in S^2−^ conversion. Hence, culture supernatants were used for further studies.

To confirm that the S^2−^ transformation was enzymatic, the culture supernatant of strain L1 was exposed to heat, SDS, PK, and SDS + PK treatment, respectively. The results are shown in Figure 3b. The conversion rate of S^2−^ did not decrease but increased by 6.04% after heating for 10 min. After heating for 1 h, the conversion rate of S^2−^ basically disappeared. The conversion rate increased after heating for 10 min, which may be due to the heat-activating effect [31], whereas heating for 1 h could lead to denaturation and aggregation of proteins due to prolonged heat, resulting in loss of enzyme activity [19]. When the CS was exposed to SDS, the conversion rate decreased by 16.92%. When exposed to PK, the S^2−^ conversion rate decreased 40.58%. When exposed to PK + SDS, the S^2−^ conversion ability was lost. This could be due to SDS disrupting the spatial structure of proteins and PK disrupting the peptide bonds between amino acids in proteins, thereby inactivating or degrading proteins [32,33]. The above results show that CS plays a major role in the conversion, and the conversion of S^2−^ is mainly catalyzed by the enzyme.

In previous studies, the conversion of S^2−^ by microorganisms was mainly owed to enzymatic catalytic actions. Such as the use of intracellular enzymes, sulfide quinone oxidoreductase (SQR) and persulfide dioxygenase (PDO), produced by heterotrophic bacteria for sulfide oxidation [34]. Flavocytochrome c sulfide dehydrogenase (FSCD), sulfur oxidase Sox multi-enzyme complex, etc., can oxidize low-valent sulfur to high-valent sulfur containing inorganic compounds [35,36].

### 3.3. Optimization of Enzyme Conversion Activity

The effects of different carbon nitrogen sources on enzyme conversion activity are shown in Figure 4. Under different carbon sources, the maximum enzyme conversion activity (63.37%) was reached by adding sucrose, and when the sucrose concentration was 5 g/L, the maximum conversion activity (66.92%) was reached. However, the difference between the conversion activity of 5 g/L and 3 g/L (66.81%) was small, based on the principle of suitability for production and economy; 3 g/L was selected as the additional concentration.

Under different nitrogen sources, the highest conversion activity (68.93%) was achieved in the presence of NH_4_Cl, at 1 g/L. Therefore, the nitrogen source of the medium was adjusted to 1 g/L NH_4_Cl. Inorganic nitrogen sources such as NH_4_Cl and NH_4_H_2_PO_4_ are suitable for S^2−^ conversion, which may be due to the easy uptake and low energy-dissipation of ammonium [37]. In summary, the most suitable carbon source is 3 g/L sucrose, and the most suitable nitrogen source is 1 g/L NH_4_Cl.

### 3.4. Extraction and Purification of Sulfur Convertase

#### 3.4.1. Extraction and Purification of Sulfur Convertase

The purification results of sulfur convertase by ammonium-sulfate-graded precipitation are shown in Figure 5a. S^2−^ conversion activity in the precipitate increased with the ammonium sulfate concentration. When the concentration of ammonium sulfate reached 60% to 80% (saturation), the activity was at its maximum. Therefore, the interval of 60% to 80% ammonium sulfate saturation was chosen to precipitate crude sulfur convertase.

Next, the enzyme was further purified by SephadexG-75 gel filtration chromatography, and four protein peaks were observed (Figure 5b). The highest specific sulfur convertase activity was detected in the second peak (50–70 min). After two purifications, the specific activity of sulfur convertase increased from 2.59 to 14.29 mg/mg, a 5.52-fold purification (Table 2).

#### 3.4.2. Protein Molecular Weight of Sulfur Convertase

The molecular weights of purified sulfur convertase were estimated using SDS-PAGE (Figure 5c,d). Compared to strains L1 and CS, the purified sulfur convertase showed a single protein band of approximately 70 kDa, indicating that the purification achieved good effects.

### 3.5. Property Studies of the Purified Sulfur Convertase

#### 3.5.1. Activity and Stability of Sulfur Convertase at Different Temperatures

From 10–80 °C, the sulfur convertase activity showed a trend of increasing, decreasing, increasing again, and then finally decreasing with increasing temperature. When the temperature reached 70 °C, the highest conversion activity was achieved (Figure 6a). The sulfur convertase can remain stable at 10–70 °C, with relative activity over 80%. The big difference in enzyme activity between the two groups at 50–60 °C may be due to the different contact times of the enzymes with the different temperatures. In previous studies, sulfite: acceptor oxidoreductase, the key enzyme for sulfide oxidation in strain LYH-3, reached maximum specific activity at 50 °C [38]; sulfur dioxygenase in *Acidithiobacillus* ferrooxidans showed the highest activity at 35 °C [39]; SQR in *Urechis unicinctus* had the highest enzyme activity at 37 °C [40]. The sulfur convertase in this study has a higher optimal reaction temperature, which may contribute to the practical application of the enzyme.

#### 3.5.2. Sulfur Convertase Stability over Storage Time

Stability over storage time is also one of the factors affecting the application of enzymes. As the storage time increases, the relative activity of the enzyme gradually decreases. On the 30th day, the conversion activity of the sulfur convertase still exceeded 50% (Figure 6b). This indicates that the enzyme has good storage stability. Previous studies have shown that low storage stability limits the practical application of enzymes [41].

#### 3.5.3. Effect of Inhibitors and Organic Solvents on Sulfur Convertase Activity

The stability of enzymes can affect their application [42]. Therefore, this section combines the application environment to study the effects of inhibitors and organic solvents on enzyme stability. The effect of inhibitors on the conversion activity is shown in Figure 6c. The three inhibitors had less effect, and the conversion activity was above 90%. Inhibitors also can be used to analyze which active residues are present in an enzyme and to classify the enzyme [43]. PMSF is a serine inhibitor and showed weak inhibition on the enzyme, indicating that the enzyme may not belong to the serine enzyme group [44]. Among the three inhibitors, NaN_3_ exhibited the highest inhibition. NaN_3_ can block the electron transfer ability of the cytochrome oxidase [45]. Therefore, this enzyme may be a member of the cytochrome oxidase family. The effect of EDTA on enzyme activity showed that low concentration inhibited and high concentration promoted, indicating that the sulfur convertase may not be a metalloprotease [46]. EDTA is a metal chelator; low concentrations inhibited the enzyme activity, which could be due to EDTA chelating the metal ions required for the conversion reaction. The promotion could be due to the high concentration of EDTA removing the impure metal ions completely, thus reducing the effect on the enzyme and promoting conversion.

Under different organic solvents, the effects of methanol, ethanol, and isopropanol on the enzyme showed the same trend, with low concentrations slightly promoted and high concentrations inhibited. Different concentrations of DMSO promoted enzymatic activity, but the promotion weakened with the increase in DMSO concentration (Figure 6d). DMSO can interact with the hydrophobic group of the protein to denature the protein [44], so it is speculated that the enzyme may not have a hydrophobic group in its active center. Overall, organic solvents have less effect on sulfur convertase activity.

#### 3.5.4. Effect of Redox on Sulfur Convertase Activity

Pyrogallic acid did not affect the relative conversion activity of the enzyme, whereas ascorbic acid, *β*-mercaptoethanol, and dithiothreitol largely inhibited the relative conversion activity of the enzyme (Figure 6e). Three reducing agents are also sulfhydryl protectors that reduce the disulfide bonds in the protein [47]. It is speculated that disulfide bonds are present in the enzyme, and they may play a key role in the sulfur conversion process.

#### 3.5.5. Effect of Zymolyte Competition on Sulfur Convertase Activity

S_2_O_3_^2−^ and SO_4_^2−^ are oxidation products of S^2−^ [35]; therefore, two substances were added to explore the effect on sulfur conversion. Different concentrations of S_2_O_3_^2−^ and SO_4_^2−^ had little effect on the conversion (Figure 6f). This indicates that the two substances may have no feedback regulation or weak regulation on S^2−^conversion.

### 3.6. Sulfur Conversion of Enzymes under Livestock Wastewater Conditions

The conversion results and the change in wastewater indicators are shown in Table 3. Previous studies have shown that bacteria can effectively remove contaminants [48], and the same results were obtained in the present experiment—strain L1 can effectively improve the conversion of sulfur, and S^2−^ was completely removed from livestock wastewater under the action of enzymes. Ammonia, total phosphorus, and COD were also reduced. Anaerobic digestion is the main technology used in wastewater treatment. In the previous studies, anaerobic digestion was used to treat slaughterhouse wastewater with 95.90% COD (600 mg/L) removal [49] and 98.4% COD (100–600 mg/L) removal for nitrogen fertilizer wastewater [50]. Compared with the current study, sulfur convertase removed more COD, but the removal rate was much lower than in the previous studies. This may be due to the different original COD of the wastewater. Compared to strain L1, sulfur convertase is more suitable for sulfur removal in wastewater conditions. Previous studies also showed that enzymatic bioremediation is superior to microbial bioremediation because it has a high bioconversion rate, a shorter remediation time, and low environmental risk [42,51]. High S^2−^ concentrations can inhibit the growth of microorganisms [8]. Moreover, H_2_S is produced at all stages of oil production, and H_2_S can be hazardous to workers’ health [52]; the rapid conversion properties of sulfur convertase may be suitable for this. In this study, the enzyme maintained stable sulfur conversion activity and also remediated wastewater, which laid the foundation for future practical application of enzymes in the environment.

## 4. Conclusions

In this study, twelve strains were isolated from wastewater; strain L1 had the highest sulfur conversion capacity, and L1 was identified as Cellulosimicrobium sp. The extracellular enzyme of the L1 showed a stronger sulfur conversion capacity; 3 g/L sucrose and 1 g/L NH_4_Cl can promote enzyme activity. After purification, the activity of sulfur convertase increased 5.52-fold; temperature, storage time, inhibitors, organic solvents, and conversion products have little effect on the enzyme, but reductants can dramatically reduce conversion activity. In wastewater conditions, the enzyme also showed stable sulfur conversion capacity. In conclusion, the current work focuses on the development and availability evaluation of the sulfur convertase, which will be identified and predicted in the future, and its mechanism of action will be studied in depth.

## Figures and Tables

**Figure 1 ijerph-19-16368-f001:**
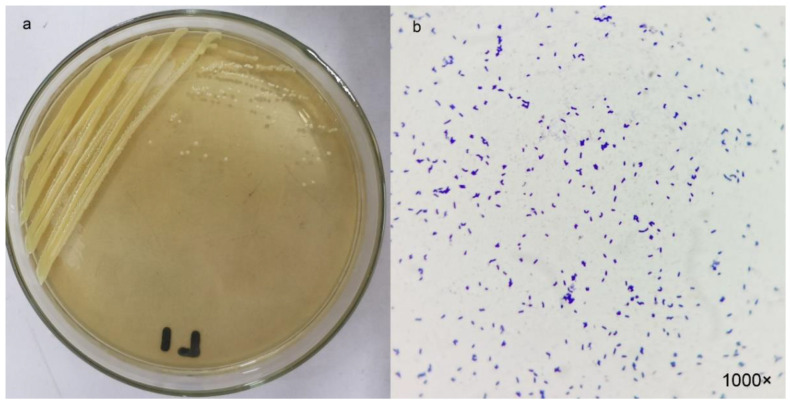
(**a**) Colony and (**b**) bacterial morphology after Gram stain of strain L1.

**Figure 2 ijerph-19-16368-f002:**
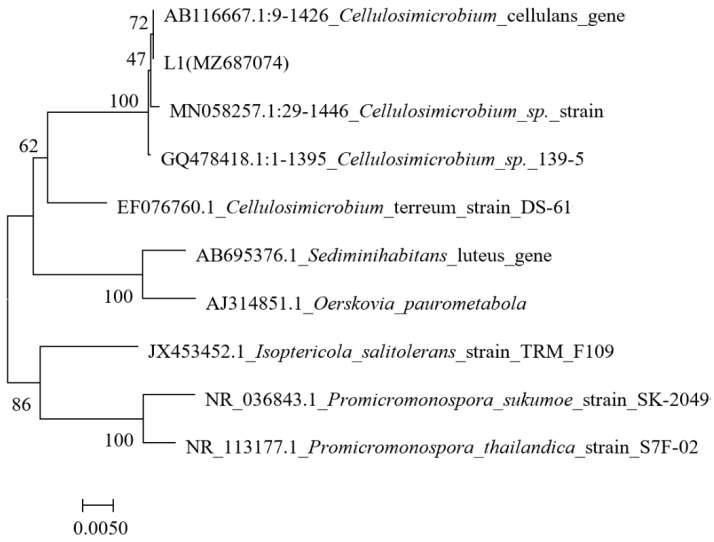
Phylogenetic tree of strain L1 (the neighbor-joining method was used); the scale bars represent 0.005 substitutions per site.

**Figure 3 ijerph-19-16368-f003:**
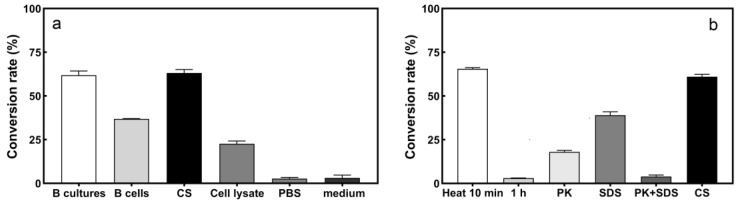
Location of active components by strain L1 and the effect of different treatments on CS. (**a**) S^2−^ conversion rate of the B cultures, B cells, CS, and cell lysate of strain L1; (**b**) the S^2−^ conversion rate of CS under different treatments.

**Figure 4 ijerph-19-16368-f004:**
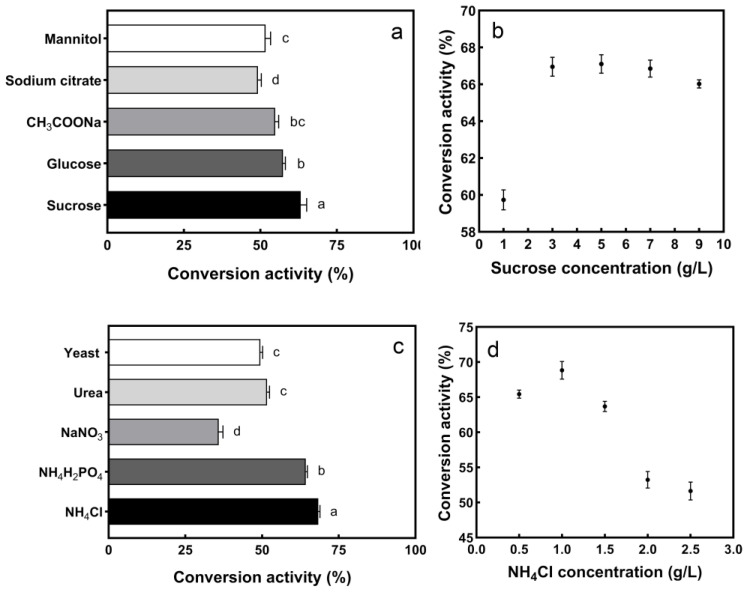
Effect of different carbon nitrogen sources on enzyme conversion activity: (**a**) carbon source, (**b**) sucrose concentration, (**c**) nitrogen source, (**d**) NH_4_Cl concentration. The different letters indicate significant differences (*p* < 0.05) between each group.

**Figure 5 ijerph-19-16368-f005:**
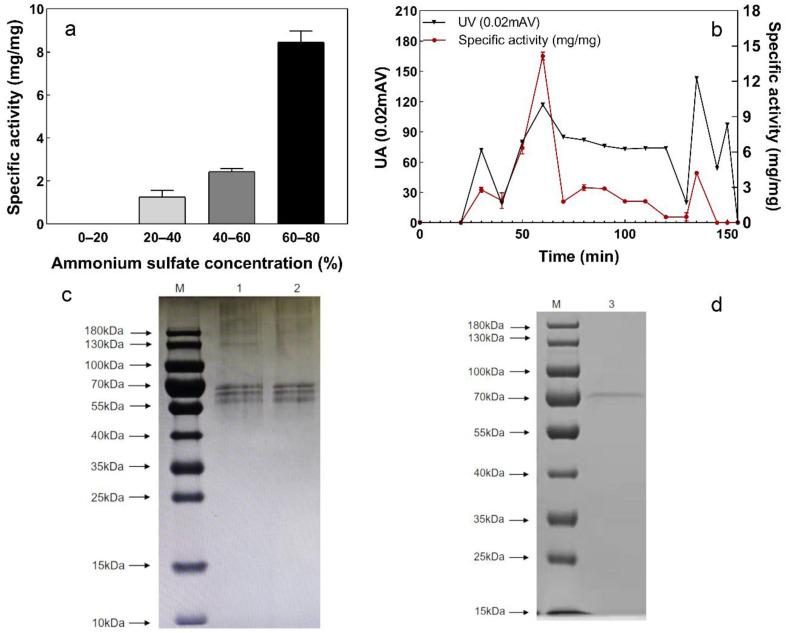
Purification of sulfur convertase, (**a**) specific activity of precipitated enzymes at different ammonium sulfate concentrations, (**b**) precipitation time and specific activity of each protein; specific activity: the amount of sulfide (mg) converted per mg of protein for 24 h, (**c**,**d**) the SDS-PAGE images of strain L1, CS, and sulfur convertase M: protein marker; 1: strain L1; 2: CS; 3: sulfur convertase.

**Figure 6 ijerph-19-16368-f006:**
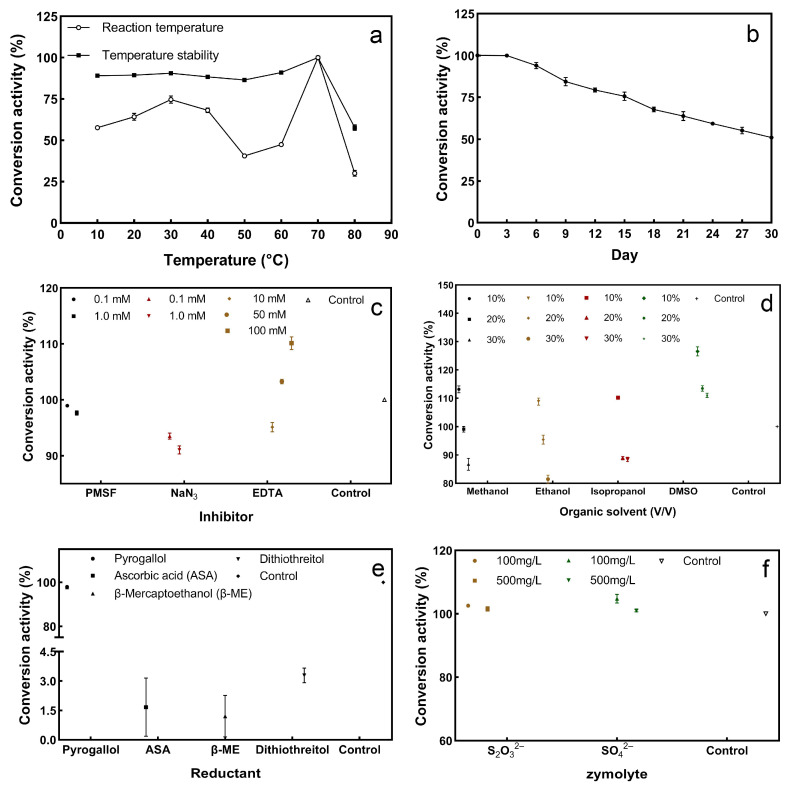
Effect of physiochemical factors on sulfur convertase activity: (**a**) the change in sulfur-converting-enzyme activity and stability at 10–80 °C; the maximum enzyme activity was set at 100%, and this was used as a basis to calculate the relative enzyme activity at the other reaction temperatures, (**b**) sulfur convertase stability over storage time, (**c**) effect of inhibitors on sulfur convertase, (**d**) effect of organic solvents on sulfur convertase, (**e**) effect of reductants on sulfur convertase, (**f**) effect of zymolytes on sulfur convertase; the conversion activity of the control group was set to 100%, and this was used as a basis to calculate the relative activity of the enzyme under inhibitors, organic solvents, reductants, and zymolytes.

**Table 1 ijerph-19-16368-t001:** S^2−^ conversion rate of different strains isolated from wastewater.

Strain	S^2−^ Conversion Rate (%)		
	24 h	48 h	72 h
L1	26.48	47.57	71.87
L2	19.45	38.42	47.03
L3	22.23	42.15	51.94
L4	20.06	31.39	37.51
L5	14.62	27.42	34.57
L6	15.74	24.46	29.14
S1	20.46	36.18	41.25
S2	7.46	19.53	23.67
S3	10.59	24.06	36.01
S4	5.09	13.46	14.92
S5	7.86	21.24	30.31
S6	30.28	43.51	62.40
CK	3.80	11.65	14.50

**Table 2 ijerph-19-16368-t002:** Steps of the purification of sulfur convertase.

Purification Steps	Total Activity (mg)	Total Protein (mg)	Specific Activity (mg/mg)	Purification (Fold)
Crude sulfur convertase	0.342	0.132	2.59	1.00
(NH_4_)_2_SO_4_ precipitation (60~80%)	0.240	0.028	8.57	3.31
Sephadex G-75	0.5	0.035	14.29	5.52

**Table 3 ijerph-19-16368-t003:** The change of wastewater indicators and S^2−^.

Group	Ammonia Nitrogen (mg/L)	Total Phosphorus (mg/L)	COD (mg/L)	S^2−^ (mg/L)
L1	1747.0 ± 54.6	196.1 ± 3.8	35002.0 ± 352.0	0.0 ± 0.0
Sulfur convertase	1565.2 ± 45.8	167.7 ± 14.9	32478.0 ± 0.0	0.0 ± 0.0
Control	1729.8 ± 3.2	296.5 ± 0.7	37196.0 ± 248.0	75.7 ± 3.1

## Data Availability

All data generated or analyzed during this study are included in this published article.

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
