# Peer review of "Extraction, Purification, Characterization and Application in Livestock Wastewater of S Sulfur Convertase"

_ijerph, 2022, doi:10.3390/ijerph192316368_

Round 1
Reviewer 1 Report
Overall
This manuscript investigated the sulfur conversion capacity of an extracellular enzyme extracted from a strain Cellulosimicrobium sp. L1 isolated form wastewater. In this study, the effects of different environmental factors on sulfur conversion transformation ability of extracellular enzyme were mainly studied and discussed. The authors elaborated the function of sulfur conversion in extracellular enzyme after purified was stability. In addition, this manuscript provides some pieces of new information regarding the performance of sulfur conversion function of enzyme in livestock wastewater. But there was insufficient discussion on its implications. The overall quality of this manuscript necessitates improvement before possible publication. There are some specific comments are given below.
Specific comments:
(1) In Table 3, it is suggested that The name of group “Self-remediation” be changed to “Control”, which can make a more obvious comparison.
(2) In Lines 334-346, the change of S2- less than 100 mg/L, but the change of COD concentration more than 4000 mg/L, the effect of functional expression of sulfur conversion is not obvious.
(3) In Lines 39-40, what was traditional microbial processes?
(4) In Figure6 (a), Could the authors add some comparison (below 30 ℃) the function of sulfur convert in enzyme at different temperatures.
(5) In section 3.5.5, S2O3- and SO42- were added to explore the effect on sulfur conversion. Interestingly, S2O3- was unstable and reduced to SO42-, resulting in a small difference in results?
(6) In figure 6 (a), Please clarify why was there a big difference between the reaction temperature and the thermal stability experiment at 50-60 ℃?
(7) What is the concentration of S2- in livestock wastewater? What is the enzyme content added into the livestock wastewater? Some details are need in M&M.
Reviewer 2 Report
In wastewater system, the main target is carbon (C) and nutrient (N and P). Recently, the harmness of sulfide in sewer network stimulated the interests of S removal in wastewater system. The authors presented an extracted S-removal enzyme from wastewater, and applied it to effectively remove S in livestock wastewater that contains really high concentrations of N, P and S. The results of S=0.0 mg/L is surprising and promising, demonstrating its potential to enhance the S removal in wastewater treatment system. Although there're some points need more explanation, the full paper is well organized, the experiments are careful designed, and the data interpretation is controled and completed.
1. Seeding of the bacteria.
I noticed that the strain L1 is extracted from wastewater that removed solid. Acutally, in wastewater, the major part of suspended solid is bacteria. Usually, activated sludge from treatmetn facilities is good as a seeding source by considering its big diversity of bacterial community. It needs explanation that why not select activated sludge, and why applied solid-liquid separation to the fresh wastewater.
2. Control of the S2- activity of the strains
I found the control set also contribute 14% of S redution in 3 days. If CK is bactrial free, the data is questionable and need more explanation that whether some physiochemical reaction happened.
3. Experiments of S removal in livestock wastewater
In case of wastewater treatment, the complicated matrix of wastewater may destroy the conversion activity of pure enzyme. I found there're statitic data in Table 3 but no number of replicates available. The reproductivity in livestock wastewater is attractive and it will be good if more information will be supplied, e.g., source of livestock wastewater, compare with the performance of typical digestion, etc.
4. Potential application fo S-convertase
Considering the cost of enzyme production, and the high dosage of 1% v/v in the designed experiments, its application in practice is quite challenged. It's better to find out some key and expensive scenarios that worth of using such convertase, e.g., rapid removal H2S in some sensitive sites.
